# Marine Derived Strategies Against Neurodegeneration

**DOI:** 10.3390/md23080315

**Published:** 2025-07-31

**Authors:** Vasileios Toulis, Gemma Marfany, Serena Mirra

**Affiliations:** 1Departament de Genètica, Microbiologia i Estadística, Facultat de Biologia, Universitat de Barcelona, 08028 Barcelona, Spain; vtoulis@ub.edu (V.T.); gmarfany@ub.edu (G.M.); 2Centro de Investigación en Red de Enfermedades Raras (CIBERER, ISCIII), Universitat de Barcelona, 08028 Barcelona, Spain; 3Institute of Biomedicine—Institut de Recerca Sant Joan de Déu (IBUB—IRSJD), Universitat de Barcelona, 08028 Barcelona, Spain; 4Stazione Zoologica Anton Dohrn, Department of Biology and Evolution of Marine Organisms, Villa Comunale, 80121 Naples, Italy

**Keywords:** marine drugs, neurodegeneration, retinal diseases, Alzheimer’s disease

## Abstract

Marine ecosystems are characterized by an immense biodiversity and represent a rich source of biological compounds with promising potential for the development of novel therapeutic drugs. This review describes the most promising marine-derived neuroprotective compounds with strong potential for the treatment of neurodegenerative disorders. We focus specifically on the retina and brain—two key components of the central nervous system—as primary targets for therapeutic interventions against neurodegeneration. Alzheimer’s disease and retinal degeneration diseases are used here as a representative model of neurodegenerative disorders, where complex molecular processes such as protein misfolding, oxidative stress, and neuroinflammation drive disease progression. We also examine gene therapy approaches inspired by marine biology, with particular attention to their application in retinal diseases, aimed at preserving or restoring photoreceptor function and vision.

## 1. Introduction

The central nervous system (CNS), including the brain, spinal cord, and retina, consists of highly specialized tissues responsible for processing and transmitting information throughout the body. In total, around 100 billion neurons and 1000 billion glial (support) cells make up the human brain [1,2].

Neurodegenerative diseases (NDDs) are extremely heterogeneous conditions characterized by the progressive loss of neuronal health [3]. These processes lead to the gradual breakdown of neural networks, synaptic dysfunction, and functional impairment [4]. Unlike many other tissues, the CNS has very limited regenerative capacity. Tissue repair is further complicated by the CNS’s intricate architecture and the hostile microenvironment triggered by neuronal cell death and neurodegenerative process [5]. These factors make neurodegenerative damage irreversible. NDDs may have genetic, environmental, and multifactorial origins, and include both common and rare diseases such as Alzheimer’s disease (AD), Parkinson’s disease (PD), multiple sclerosis, age-related macular degeneration (AMD), and retinitis pigmentosa (RP). Many NDDs have a devastating outcome in people across all age groups. Currently, there is no cure for most NDDs, and available treatments only alleviate symptoms or slow the disease progression, with a significant economic and emotional cost [6]. Ongoing research explores several strategies, including gene-therapy and stem cell therapy approaches, to halt or reverse neurodegeneration [7,8]. However, a cure for NDDs remains an important and unmet medical need.

According to the World Register of Marine Species (WoRMS), there are over 247,898 accepted marine species, thriving in highly diverse and often extreme environments conditions (ranging in light, pressure, oxygen, and temperature) that are inhospitable to most terrestrial organisms. Over time, evolutionary pressures have driven the development of unique physiological, biochemical, and molecular adaptations in marine life [9]. Therefore, marine organisms offer a diverse array of bioactive compounds due to their unique evolutionary adaptations. These compounds offer potential therapeutic benefits, including anti-inflammatory, antioxidant, and neuroprotective effects. Therefore, marine-derived drugs have gained significant attention as a promising source for novel treatments aimed to fight against neurodegeneration [10].

This review highlights the increasing interest in bioactive molecules—and molecular strategies—derived from marine organisms to enhance neuronal function and health. We will focus on both retina and brain as key organs belonging to CNS and targeted by therapies against neurodegeneration. Moreover, as the number of neurodegenerative diseases affecting the brain and the research on the field are very extended, we will focus on AD. AD is the most common cause of dementia in an aging population, and it is characterized by the progressive loss of neurons and synapses in the brain, particularly in areas responsible for memory and cognition. AD exemplifies how complex molecular mechanisms, such as protein misfolding, oxidative stress, and neuroinflammation, converge in neurodegenerative disease progression, thus representing a prime example of a neurodegenerative disorder. Importantly, we will pay attention not only to marine organism-derived drugs, but also to gene therapies inspired by marine organisms, particularly employed in retinal diseases to improve or restore photoreceptor function and vision.

## 2. Neurodegeneration in Central Nervous System

NDDs may be triggered by different genetic or external stimuli, but all of them end up with the breakdown of the structure and function of neural networks and progressive loss of vulnerable neurons, eventually leading to impaired cognition, memory, behavior as well as sensory and motor dysfunction.

### 2.1. Hallmarks of Neurodegeneration

The combination of genetics, environmental exposures, and lifestyle factors all contribute to different stress thresholds for the diverse neuronal populations of the CNS, leading to a differentially affected CNS region over others in NDDs. Despite these phenotypic differences, there are relevant similarities at the molecular and cellular level. The so-called hallmarks of neurodegeneration are often explained as specific traits within each disease, but they are interrelated and can be grouped in categories in a holistic view from the molecules to the organ: from the deregulation of genetic information, loss of proteostasis, and larger organelle dysfunction to the alterations in interneuron communication and chronic neuroinflammation. Since NDDs are driven by a combination of NDD hallmarks, only multi-targeted therapies may be effective in the long run.

#### 2.1.1. Deregulation of Genetic Information

DNA damage accumulation and RNA metabolism defects are key contributors to NDDs, especially in the aging CNS. Reactive oxygen species (ROS) from mitochondrial activity induce genomic stress in non-dividing neurons, leading to mutations, chromosomal instability, and impaired transcription. To maintain function, neurons rely on DNA repair and proper RNA processing, but these systems often decline with age or are disrupted by genetic mutations, promoting disease progression [11]. Rare neurological disorders caused by mutations in genes regulating nucleic acid metabolism can lead to severe neurodegeneration without cancer predisposition. Additionally, stress granules—structures that transiently block translation under stress—can become dysfunctional due to mutations in RNA-binding proteins, disrupting mRNA dynamics and contributing to neuronal damage. Last but not least, many NDDs show misregulation of miRNAs that control the expression of relevant CNS development and maintenance genes. Although miRNAs appear as interesting key targets for therapeutics in NDDs, the fact that they regulate many gene targets not always in the same direction, hampers the use and design of effective miRNA therapeutics [12].

Epigenetic regulation, which is dynamic and reversible in nature, determines transcriptomic outcomes. Histone modifications, DNA methylation, and noncoding RNAs determine the 3D chromatin landscape and accessibility to transcription factors and the enzymes in charge of these modifications respond to specific intracellular metabolic and paracrine cues. Disease-related genetic variants altering epigenomic attributes are potentially underestimated in NDDs, but it is well-known that dysfunctional metabolism and altered epigenomic dynamics correlate with advanced age, which can impact in neurodegeneration [13]. Therefore, the epigenome–metabolism axis is currently a focus of interest for NDDs and potential therapeutic strategies.

#### 2.1.2. Loss of Proteostasis

Many NDD genes are characterized by the formation of insoluble and non-degraded specific protein aggregates (e.g., beta-amyloid Aβ_40–42_ in AD, α-synuclein in PD, phosphorylated tau neurofibrilles in fronto-temporal dementia and other tauopathies), which often are the main marker for diagnosis and disease classification [14,15]. As aggregates in NDDs are clearly associated with mutations in the genes encoding the insoluble misfolded protein, the causative mutations are considered to result in a toxic gain-of-function that finally leads to neuronal death. Therefore, protein degradation systems play an important role in the clearance of toxic protein aggregates, the two major cell protein degradation mechanisms being the ubiquitin–proteosome system (UPS) and the autophagy-lysosomal pathway (ALP) [15,16].

In NDDs, the activity of these pathways is overcome by toxic aggregates, but also, these pathways have been shown to decline with age, thus contributing to the accumulation of pathological or insoluble proteins. There is a compensatory cross-talk between UPS and ALP, although with some specialization in their activities: since the UPS mostly contributes to the degradation and turnover of oxidized, misfolded or dysfunctional proteins via proteasome, whereas the ALP is mainly in charge of degrading aggregated proteins and damaged organelles through the formation of autophagosomes and subsequent fusion with lysosomes to form autolysosomes [17]. Mutations in the genes encoding key components of the UPS and ALP cause NDDs, for instance, mutations in ubiquitin ligases or deubiquitinating enzymes [18,19]. Also many lysosomal storage disorders, where the digestion of specific membrane components by autophagolysosomes is impaired (such as Niemann-Pick, Tay-Sacs of Gaucher diseases) are characterized by neurodegenerative neuropathies [20]. Disrupted mitophagy also alters mitochondrial dynamics and metabolism. Collectively, disrupted proteostasis negatively impacts many CNS cell functions and it is considered a characteristic NDD hallmark. Therefore, therapies designed to regulate protein degradation may positively impact in neuronal cell homeostasis [21,22].

The neuronal cytoskeleton, composed of microtubules, neurofilaments, and actin filaments is essential for maintaining neuronal architecture and enabling intracellular transport. It supports key processes such as organelle trafficking, synaptic function, and ciliary signaling, all vital for neurotransmission and neuronal stress responses. In NDDs, cytoskeletal dysfunction disrupts these processes, leading to impaired signal transmission, mitochondrial transport defects, and axonal damage [23]. These abnormalities are also strongly interrelated with other hallmarks of both NDDs and aging, and contribute significantly to neurodegenerative progression [24]. Therefore, cytoskeletal abnormalities constitute a key therapeutic target for treating neurodegeneration.

#### 2.1.3. Large Organelle Dysfunction

Mitochondria are crucial organelles that provide energy to sustain cell metabolism in neurons, the highest energy demanding cells. Reduced mitochondrial function and ATP synthesis is associated with lower production of the scavenger glutathione (GSH), potentially worsening the oxidative environment of the cell [25]. Also, during aging, exposure to environmental stressors and excessive levels of the by-product ROS could lead to constitutive changes in mitochondrial DNA mutagenesis and ATP production and ROS generation, all of which are signals of mitochondrial dysfunction and cause premature cell death in the aged brain [26,27,28]. Several genes that regulate mitochondrial fusion and fission dynamics are mutated in rare genetic NDDs. Moreover, mitochondrial dysfunction also occurs in several NDDs with non-mitochondrial etiology. For instance, aggregated proteins and altered cytoskeleton can adversely impact mitochondrial transport, thus indicating that mitochondrial function, quality control, or transport are adversely affected in NDDs [29]. Beyond their function in bioenergetics, mitochondria are also involved in many cell metabolism pathways, including lipid biosynthesis, intracellular calcium homeostasis, myelin repair, and regulation of apoptosis [30]. Metabolic disturbances in the body capacity to process macronutrients such as fat, protein, and carbohydrate contribute to the aging process and are linked to NDDs. In fact, obesity is a risk factor for some NDDs and the majority of AD cases have type 2 diabetes mellitus, which associates NDD to impaired glucose metabolism [31]. NDD patients also show gut dysbiosis as a most frequent symptom, overall, linking NDDs to a broad range of metabolic disorders that could be targeted by specific compounds [32].

Oxidative and endoplasmic reticulum (ER) stress pathways induce neuronal apoptosis through a complex cellular signaling network. As the ER is mainly composed of lipid membranes and forms part of an active network with mitochondria, the production of ROS by the latter induces ER stress both by lipid peroxidation and disrupted protein folding. Both NDDs and aging are related to dysfunctional ER and an increase in cell death triggered by oxidative stress [11].

#### 2.1.4. Altered Intercellular Communication

Synaptic dysfunction and hyperexcitation or excitotoxicity are closely linked to defective energy metabolism and axon transport, as well as impaired protein homeostasis and cytoskeletal dynamics [33]. Excitotoxicity by excess of glutamate neurotransmitter causes neuronal death by excessive Ca^2+^ influx. Glutamate-mediated excitotoxicity has been considered an important mechanism in the etiology and progression in most NDDs [34]. Functional neuronal networks require precise synaptic function and a finely tuned regulation of synapse stabilization and elimination. Synaptic function is modulated by neurotransmitters, calcium changes, cytoskeletal adaptations, presynaptic and post-synaptic vesicles dynamics and signaling, all of which are highly dependent on the correct function of mitochondria and membrane pumps. Observing synaptic failure and toxicity as signals of a perturbation of specific neuronal networks has been described as an early event preceding neuronal loss in NDDs, thus indicating key therapeutic targets for neurotransmission modulation prior to the irreversible neuronal cell death occurs [35]. A successful example of this strategy is the replacement of the lost dopamine signal in PD using precursor L-DOPA, which is temporally effective at restoring motor symptoms in early to moderate disease [36].

#### 2.1.5. Organ Alterations

The immune system plays a vital role in maintaining CNS homeostasis, but chronic inflammation triggers the inappropriate activation of immunity and overwhelms the system [37]. In the CNS, microglia and astrocytes are the primary immune cells. Normally, microglia monitor the environment, remodel synapses, and phagocytose dead cells and debris. However, in response to injury or protein aggregates, they activate astrocytes and release pro-inflammatory cytokines. Sustained neuroinflammation causes neuronal damage and contributes to NDDs [38]. This inflammation, often triggered by aging, genetic mutations, or environmental stimuli, may affect initially a specific region of the CNS and progressively spread, worsening disease. Actions addressed to diminish the neuroinflammation can alleviate the NDD symptoms [39].

Abnormal angiogenesis (both excessive and insufficient) is not traditionally listed among the classic hallmarks of neurodegenerative diseases. However, recent research has revealed that impaired angiogenesis and vascular dysfunction may play a key role in the pathogenesis of several NNDs, such as AD and PD, or AMD and diabetic retinopathy [40,41]. Angiogenesis is the process of new blood vessel formation from existing vasculature [42]. It is tightly regulated by a balance of pro-angiogenic factors, such as the Vascular Endothelial Growth Factor (VEGF) and anti-angiogenic factors, such as thrombospondin and endostatin. Many physiological consequences of abnormal angiogenesis are shared by brain and retina and include VEGF dysregulation, endothelial cell dysfunction, hypoxia and oxidative stress, and neuroinflammation [43]. Importantly, impaired angiogenesis intersects with established hallmarks like inflammation, protein aggregation, and mitochondrial dysfunction.

Progressive neuronal death is another key feature of NDDs. Neurons, being non-dividing and energy-demanding cells, are vulnerable to accumulated damage and oxidative stress. Their complex structure and reliance on glial support make them especially susceptible to cell death, a characteristic that is further exacerbated by an age-related decline in resilience mechanisms [44]. Neuronal death comes in different flavors and colors, through mechanisms including apoptosis, necrosis, necroptosis, ferroptosis, pyroptosis, and others [45]. These pathways may act individually or in combination, depending on the cell type and stressors involved. Shedding light onto the key molecular players and cell death pathways is crucial to identify therapeutic compounds to treat genetic and age-related NDDs [46]. In conclusion, targeting neuroinflammation, angiogenesis, and cell death pathways holds promise for treating both genetic and age-related neurodegeneration.

### 2.2. Retina and Brain: Two Elements of the CNS

Although peripherally located, the neuroretina is an integral part of the CNS and is a direct extension of the brain [47]. The retina functions as a complex neural circuit that transforms light stimuli into action potentials, which are transmitted to the brain enabling the perception of light, color, and detailed images. Adjacent to the neural retina lies the retinal pigment epithelium (RPE), a single, non-stratified epithelial layer that is closely associated with photoreceptors and is essential for retinal homeostasis [48]. The complex network of retinal cells is composed of five primary neuronal types: photoreceptors, horizontal cells, bipolar cells, amacrine cells, and retinal ganglion cells (RGCs) [49]. These cells are organized into three distinct layers: (1) the outer nuclear layer (ONL) hosting the photoreceptor cell bodies; (2) the inner nuclear layer (INL) hosting horizontal, bipolar, and amacrine cells; (3) the ganglion cell layer (GCL) hosting the RGCs. The synaptic connections between retinal neurons form the outer plexiform layer (OPL) and inner plexiform layer (IPL). Importantly, there are two main types of photoreceptors—rods and cones—which differ in shape, expressed photopigments, distribution, and synaptic connectivity [50]. These specializations allow for distinct visual functions: the rod system is highly sensitive to light but offers low spatial resolution, while the cone system enables high spatial resolution and color vision, functioning best under high light intensity due to the differential sensitivity of cone opsins to specific wavelengths [51].

Photoreceptors perform the process of phototransduction, which is the conversion of light energy into electrical signals [52]. The resulting signal is transmitted via the synapse between photoreceptors and bipolar cells to the retinal ganglion cells, which relay the information to the brain via the optic nerve. Retinal neurons closely resemble those found in other CNS regions, such as the cortex and hippocampus. For example, retinal ganglion cells (RGCs) exhibit typical neuronal structures (dendrites, soma, and a large axon) making them relevant to broader neurological studies. Among other similarities between the retina and the brain, the inner blood–retinal barrier (BRB) shares structural and functional similarities with the blood–brain barrier (BBB), including tight junctions between cells forming the vascular endothelium, supported by Müller and astroglial endfeet [53,54]. These barriers are fundamental to restrict the penetration of molecules (e.g., neurotoxins) and cells (e.g., immune cells or infectious agents) from peripheral parts of the body into the CNS [55]. Moreover, both retina and brain CNS are able to respond to injuries triggering neuroinflammation through a variety of cell types such as astrocytes, microglia, endothelial cells, and T cells in the brain, and microglia, Müller Cells, endothelial cells and RPE cells in the retina [56]. The retina and brain share many common features, thus much information obtained from the retina could be applicable to the brain and spinal cord, and vice versa [57].

Importantly, many NDDs such as AD, multiple sclerosis, spinocerebellar ataxia, and PD, manifest with ocular symptoms that can precede brain-related signs. As an example, AD also impacts the retina, where similar pathological hallmarks (e.g., amyloid plaques and tau protein aggregation) have been observed and often precede brain-related clinical symptoms [57,58,59]. This highlights the well-established concept of the retina as a “window to the brain” and opens new avenues for early diagnosis and therapeutic intervention through retinal imaging and targeted neuroprotective strategies.

## 3. Retinal Neurodegeneration

Retinal degeneration diseases (RDDs) form a very heterogeneous group of neurodegenerative disabling diseases, including AMD, glaucoma, and diabetic retinopathy. They also include inherited retinal dystrophies (IRDs), rare diseases caused by genetic mutation(s) in genes involved in the development and function of retinal cells and affecting 1:3000 people worldwide [60]. IRDs can affect both the outer retina (namely photoreceptors, such as in RP, Leber congenital amaurosis, and macular dystrophies) and the inner retina (mainly RGCs, such as in Leber’s hereditary optic neuropathy and dominant optic atrophy). The process initiates with the attrition of retina integrity followed by the disintegration of retinal cell lining and the death of photoreceptors. Once initiated, retinal degeneration is irreversible and leads to blindness. Despite that, there is no curative treatment for most RDDs [61]. Although the prevalence, etiology, pathogenesis, and clinical characteristics of RDDs are very different, the classical common hallmarks of neurodegeneration can be identified, which include neuroinflammation, microglia activation, and mitochondrial dysfunction [62]. A growing body of research has demonstrated the neuroprotective properties of marine natural products in the context of RDDs: several studies underscored the potential of marine-derived compounds in preventing or delaying the progression of ocular diseases [63]. Importantly, biomedical strategies based on molecular solutions inspired by marine organisms also show a huge potential in treating retinal degeneration. An evident paradigm of this potential is represented by aquatic microbial opsins used in optogenetics to activate neurons engineered by ectopically expressing a light-sensitive protein delivered by Adeno-associated Viral (AAV) vectors. This approach aims to at least partially restore neuromodulation and electrophysiological activity of the damaged neuroretina in case of advanced retinal degeneration [50].

### 3.1. Marine Drugs in Retinal Neurodegeneration

Marine-derived molecules show promise in fighting retinal neurodegeneration by offering neuroprotective, antioxidant, and anti-inflammatory effects. Here, we provide an exhaustive overview of marine-derived compounds with potential for retinal neurodegenerative diseases (Figure 1 Table 1). The molecular structure is presented in Figure 2.

#### 3.1.1. Marine Caroteinods

A promising class of molecules with therapeutic potential in RDDs are marine carotenoids, lipophilic pigments produced by microalgae, seaweed, and marine animals such as krill or salmon, assuming them through diet. Carotenoids serve multiple vital roles in marine organisms, including protecting chlorophyll by absorbing excess light energy and scavenging ROS. Despite their valuable bioactive properties, carotenoids present several limitations as bioactive molecules to use at a pharmacological level, e.g., the susceptibility to degradation, short shelf life, poor water solubility, and low bioavailability.

Marine carotenoids with relevant pharmacological activities in retinal degeneration are fucoxanthin and astaxanthin. Fucoxanthin is an orange carotenoid found in several brown seaweeds such as *Hizikia fusiformis*, *Laminaria japonica*, and *Sargassum fulvellum*, extensively studied for its multifaceted bioactivities. It represents a promising candidate for therapeutic application in retinal neurodegenerative diseases, particularly AMD. In a study by Liu et al., fucoxanthin demonstrated protective effects in both in vitro (in ARPE-19 cells) and in vivo (photo-induced retinal damage) models [64]. In ARPE-19 cells, fucoxanthin suppressed the overexpression of VEGF, a key mediator in pathological angiogenesis. Additionally, the compound improved phagocytic function and enhanced ROS clearance, suggesting improved cellular resilience against oxidative stress [64]. In vivo experiments confirmed that fucoxanthin provides significant protection to the retina against oxidative stress-induced damage; it has been studied in a sodium iodate (NaIO_3_)-induced AMD animal model, where fucoxanthin administration reduced photoreceptor’s death and protected RPE cells from oxidative stress injury [65]. In ARPE-19 cells treated with hydrogen peroxide, fucoxanthin reduced ROS generation and malondialdehyde (MDA) levels—a marker of lipid peroxidation—and increased the mitochondrial metabolic rate [65]. These results highlight the therapeutic potential of fucoxanthin in retinal neurodegenerative diseases through mechanisms involving the preservation of mitochondrial function, protection from oxidative damage and anti-angiogenic modulation.

Astaxanthin is another carotenoid pigment, well known for its potent antioxidant and anti-inflammatory properties. It is widely produced by marine microorganisms such the marine algae *Haematococcus pluvialis* and the red yeast *Xanthophyllomyces dendrorhous*. The pharmacological potential of astaxanthin has been studied in multiples fields—including cardiovascular diseases, metabolic syndrome, gastric ulcers, and cancer—as anti-inflammatory and antioxidant factors. It can cross the blood–brain barrier, making it particularly relevant for neurological health and has been already commercially available as a dietary supplement [90]. In the retina, astaxanthin exhibits protective effects against light-induced damage, primarily through its antioxidant activity, in both in vivo and in vitro models [91]. Astaxanthin was reported to reduce the apoptosis of RGCs in diabetic *db/db* mouse—model of type-2 diabetes—and decrease the levels of oxidative stress markers. In addition, astaxanthin attenuated hydrogen peroxide-induced apoptosis in the rat retinal ganglion cell line RGC-5 [66].

Oral administration of astaxanthin has neuroprotective effects and reduces ocular oxidative stress and inflammation in streptozotocin (STZ)-induced diabetic rats, preserving retinal function and protecting against oxidative damage which may be mediated by downregulation of NF-κB activity [67]. In agreement with these results, mouse photoreceptor cells (661W) cultured in high-glucose media and treated with different concentrations of astaxanthin showed a dose-dependent reduction in ROS production and apoptosis, an effect mediated by the PI3K/Akt/Nrf2 pathway [68]. Astaxanthin has also been described as a neuroprotector in mouse models of acute glaucoma. Indeed, the apoptosis of RGCs induced by ischemia and reperfusion is repressed in astaxanthin-treated mice, through a molecular mechanism involving the Nrf2/Ho-1 pathway [69].

#### 3.1.2. Diphlorethohydroxycarmalol

Diphlorethohydroxycarmalol (DPHC) is a phlorotannin isolated from the brown alga *Ishige okamurae*, known for its potent antioxidant, anti-inflammatory, and anti-angiogenic properties. Recent studies highlighted its potential therapeutic applications in retinal neurodegenerative conditions, such as AMD [70]. DPHC protected RPE human cell line ARPE-19 cells against H_2_O_2_-induced DNA damage and apoptosis by scavenging ROS and inhibiting the mitochondrial-dependent apoptotic pathway [70].

#### 3.1.3. Fucoidan

Fucoidan is a sulfated polysaccharide primarily found in various species of brown seaweed and most commonly extracted from *Fucus vesiculosus* and *Undaria pinnatifida*. It represents a promising drug for several retinal diseases, particularly AMD, for its antioxidant, anti-coagulant, anti-thrombotic, immunoregulatory, anti-viral, and anti-inflammatory effects [92]. Fucoidan reduces secretion and expression of VEGF in the retinal pigment epithelium and reduces angiogenesis in vitro [71].

Fucoidan also has a protective effect against the epithelial–mesenchymal transition (EMT) of RPE cells and the progression of proliferative vitreoretinopathy (PVR) which is linked to retinal detachment, probably due to EMT of RPE [72]. In RPE, fucoidan was able to reverse the transforming growth factor (TGF)-β1-induced EMT, to regulate the expression of α-smooth muscle actin (α-SMA), fibronectin, E-cadherin, and phosphorylated Smad2/3, as well as to inhibit the migration and contraction of these cells. In in vivo models of PVR, fucoidan arrested the progression of PVR in the eyes and suppressed the formation of α-SMA-positive epiretinal membranes [72]. Further studies demonstrated that low molecular weight fucoidan (LMWF) extract from brown algae alleviates diabetic retinal neovascularization and damage in streptozotocin-induced diabetic mice, via inhibition of hypoxia-inducible factor-1α and VEGF, providing a candidate drug for prevention and treatment of diabetic retinopathy [73]. More recently, fucoidan has been described as a potential drug in hypertensive retinopathy by mitigating angiotensin II-induced dysfunction via modulating the expression of Sirtuin1 and the NOD-like receptor thermal protein domain associated protein 3 (NLRP3), both in in vivo and in vitro models [74].

#### 3.1.4. 4-(Phenylsulfanyl)butan-2-One

4-(Phenylsulfanyl)butan-2-one (4-PSB-2) is a synthetic derivative of the marine natural product austrasulfone, originally isolated from the soft coral *Cladiella australis*. This compound has been investigated for its potential therapeutic effects in retinal degeneration, particularly in AMD and optic nerve injury models [75]. In RPE cells exposed to amyloid-β oligomers, 4-PSB-2 can reduce the expression ofTNF-α, COX-2 and iNOS via NF-κB inhibition, indicating its potential to modulate inflammatory pathways [75]. Administration of 4-PSB-2 in rat models of optic nerve injury led to increased RGC survival, due to an anti-inflammatory and anti-apoptosis action. The 4-PSB-2 treatment reduced iNOS and COX-2 expression, decreasing retinal inflammation and increasing RGC survival and visual function [76].

#### 3.1.5. Homotaurine

Homotaurine, a taurine analog, has garnered attention for its neuroprotective properties in retinal degeneration, particularly in conditions like glaucoma and diabetic retinopathy [93]. It is naturally present in marine red algae like *Hypnea boergesenii*, *Gracilaria corticate*, and *Gracilaria pygmae*. While research is ongoing, several studies highlight its potential therapeutic effects. Davinelli et al. showed that co-treatment with citicoline improves cell survival in primary retinal cultures under experimental conditions simulating retinal neurodegeneration [77]. In rodent models of hypertensive glaucoma, dietary supplementation with a combination of homotaurine, forskolin, spearmint extract, and B vitamins has been shown to protect RGCs. This protection is associated with a reduction in calpain activity and decreased levels of the apoptotic markers caspase-3 and PARP-1, indicating that homotaurine contributes to RGC survival through anti-apoptotic mechanisms [78,79,80].

Importantly, clinical trials have been performed to evaluate the potential beneficial and synergistic effects of oral intake of a fixed combination of citicoline plus homotaurine (CIT/HOMO) on RGC function in subjects with glaucoma. Administration of CIT/HOMO for 4 months improved the function of inner retinal cells recorded by using pattern electroretinogram (PERG), showing positive effects on neuromodulation [81].

#### 3.1.6. Largazole

Largazole, a marine-derived cyclodepsipeptide isolated from the cyanobacterium *Symploca hydnoides*, has gathered attention for its potential therapeutic effects in retinal neurodegeneration disorders for its anti-inflammatory and anti-angiogenetic properties. While direct studies on largazole’s impact on retinal diseases are limited, its known mechanisms of action in other pathological contexts suggest promising applications. Indeed, largazole inhibits angiogenesis by selectively acting on class I histone deacetylases (HDACs) [82,83].

#### 3.1.7. Heparin-like Compound

Recently, a heparin-like compound isolated from marine shrimp *Litopenaeus vannamei* presented anticoagulant and hemorrhagic potential, reducing acute inflammation in animal models [84]. Moreover, it has been suggested as a candidate drug for treating neovascular AMD and other angioproliferative diseases, as it showed antiangiogenic and anti-inflammatory activities, both in vitro (treatment in ARPE-19) and in vivo (intravitrous administration in laser-induced choroidal neovascularization murine models). This compound binds to growth factors (FGF-2, EGF and VEGF), blocks endothelial cell proliferation and induces the decrease in VEGF and TGF-β1 expression levels in the choroidal tissue. The final effect is the reduction in the choroidal neovascularization area [85].

#### 3.1.8. Spirulina

Although not a compound, *Spirulina* is an undifferentiated spiral-shaped cyanobacterium (blue–green alga) that grows in alkaline and saline environments and mainly consists of two species: *Spirulina platensis*, and *Spirulina maxima*. Spirulina is cultivated worldwide, is suitable for human consumption and is used as a nutritional supplement, being a source of valuable protein, polyunsaturated fatty acids, and antioxidants. Spirulina has shown potential benefits in managing retinal degeneration, primarily due to its content in bioactive compounds that may be beneficial for retinal health, such as the antioxidant phycocyanin, beta-carotene, zeaxanthin, vitamin E, C, and B12. Protective effects of *Spirulina maxima* against blue light-induced retinal damages were shown both in vitro in A2E-Laden ARPE-19 cells and in vivo in Balb/c mice [86]. These findings were confirmed by further studies investigating whether a 4 week long spirulina-supplemented diet (20% Spirulina), would suppress photo stress-induced retinal damage and prevent vision loss in BALB/cAJcl mice [87]. After acute luminic stress (3000-lux white light for 1 h), visual function was analyzed by immunohistochemistry and electroretinography. Authors showed that spirulina supplementation ameliorates the thinning of the photoreceptor layer, reduces the glial fibrillary acidic protein activation as well as photoreceptor cell death [87]. At the molecular level, spirulina uptake reduces retinal ROS levels, superoxide dismutase 2 and heme oxygenase-1 expression, while it activates Nrf2 expression. Globally the authors conclude that spirulina supplementation protects retinal neurons from oxidative (luminic) stress [87].

#### 3.1.9. Wondonins

Wondonins are marine alkaloids produced by the sponges *Poecillatra wondoensis* and *Jaspis* sp. [88]. The therapeutic potential of wondinins in retinal neurodegeneration has been tested in diabetic retinopathy, a retinal complication linked to excessive angiogenesis. Targeted chemical transformations of wondonin produced a new compound with minimized cytotoxicity and antiangiogenic activity, which has been tested in a zebrafish model of diabetic retinopathy [88]. The compound was further chemically modified, generating compound 31 with enhanced solubility, more bioactivity, and less cytotoxicity. Molecular studies revealed that compound 31 suppressed high glucose–induced angiopoietin-2 (ANGPT2) expression in retinal cells. Preclinical studies performed in mouse models of choroidal neovascularization and oxygen-induced retinopathy confirmed the antiangiogenic activity of compound 31, highlighting its promising features for developing antiangiogenic small molecule therapies for retinal neurodegeneration involving aberrant neovascularization [89].

### 3.2. Microbial Opsins in Optogenetics

Optogenetic vision restoration is a revolutionary therapeutic approach that enables precise control of neural activity using light-sensitive proteins ectopically expressed in genetically modified cells and offering a powerful tool for neuroscience [94,95]. Optogenetics can be used in modulating dysfunctional brain circuit, offering potential treatments for PD, epilepsy, depression, and other neurological disorders [96]. However, one of the most promising applications of optogenetics is the treatment of retinal degenerative diseases, particularly RP, which is a leading cause of inherited blindness affecting about 1 in 4000 people worldwide [97]. RP involves progressive photoreceptor loss that leads to structural and functional changes in the retina, including abnormal synaptic connections and increased spontaneous activity, which also affects the visual cortex. However, the inner retinal cells like bipolar and retinal ganglion cells often remain intact, making them suitable targets for optogenetic therapy [98]. Optogenetic therapy aims to convert these surviving cells into artificial photoreceptors by introducing light-sensitive proteins. This approach offers a gene/mutation-independent solution for advanced stages of RP, as it can restore light sensitivity even when photoreceptors are lost. Adeno-associated Virus (AAV) represents the most common delivery vectors for opsins in optogenetics-based gene therapy. Following opsin transduction, target cells become light-sensitive, effectively acting as surrogate photoreceptors.

Both microbial and animal opsins have been selected as light-sensitive protein to target retinal cells via optogenetics. Importantly, microbial opsins are the molecular foundation of optogenetics, as they have been used in the first clinical trials in RP patients (NCT02556736; NCT03326336). The microbial opsins can be categorized into two types: depolarizing and hyperpolarizing. Depolarizing opsins are light-gated ion channels that upon light activation allow the entry of cations inside the cell, thus causing depolarization. These proteins mimic ON responses in bipolar cells and RGCs. Hyperpolarizing opsins are light-gated chloride pumps that inhibit neurons by hyperpolarization in response to yellow or green light stimulation. They are set for OFF responses in bipolar cells or reactivating dormant cones [99].

#### 3.2.1. Depolarizing Opsins

The first optogenetic therapeutic approach used Channelrhodopsin-2 (ChR2), a light-gated ion channel from *Chlamydomonas reinhardtii*. This microbial opsin restores light responses in RGCs and bipolar cells in *rd1* mice and RCS rats, but requires intense blue light and is ineffective beyond 550 nm [100,101,102,103]. The ChR2 mutant CatCh (Calcium translocating channelrhodopsin) presents an enhanced calcium permeability and 70 × greater light sensitivity. It is effective in macaques but still needs relatively strong light [104,105]. PsCatCh2.0 is another advanced optogenetic tool derived from a channelrhodopsin originally isolated from *Platymonas subcordiformis* (PsChR). It is notable for its blue-shifted activation, enhanced ion conductance with improved calcium, sodium permeability, superior light sensitivity, and fast kinetics. It is also optimized for expression in RGCs, making it suitable for late-stage retinal degeneration. Preclinical studies in blind rodent models confirmed its functional integration into the visual pathway [106]. CoChR-3M has been developed from the mutagenesis of ChR from the green alga *Chloromonas oogama* for ambient light sensitivity and used successfully in triple-knockout mice [107]. Chronos (ShChR) and ChronosFP are channelrhodopsins derived from *Stigeoclonium helveticum* and show fast kinetics and high sensitivity. Both preclinical studies [108,109] and ongoing clinical trials (ID NCT04278131) pointed Chronos as an essential and safe opsin for clinical research.

Different from the Channelrhodopsins—classic blue-light sensitive tools for optogenetic activation—red-shifted opsins are engineered or naturally occurring microbial opsins that respond to longer wavelengths of light, typically orange, red, or even near-infrared (600–700+ nm). Red-shifted variants present deeper tissue penetration, allowing activation of deeper cells with less invasiveness and reduced phototoxicity. Moreover, they can enable simultaneous control of multiple cell types with different opsins activated by distinct light colors. Here, we report an example of red-shiftedopsins: VChR1 from *Volvox carteri* could effectively stimulate neurons in wavelengths less harmful to retinal cells [110]. However, it shows poor membrane trafficking in mammals [111]. ReaChR is a red-activatable ChR, engineered through mutagenesis of VChR1 to improve red-light sensitivity (>600 nm), membrane localization, and trafficking [112]. It is effective in mice, macaques, and human retinal explants, restoring light sensitivity with functional cortical and behavioral responses [113]. bReaChES is a variant of ReaChR with further enhanced light sensitivity and temporal precision. Its use in mouse models has shown functional improvements [114].

ChrimsonR is a genetically engineered variant of the natural channelrhodopsin called Chrimson, originally identified in the alga *Chlamydomonas noctigama*. It is the most red-shifted opsin to date, with a peak activation around 590–595 nm. Preclinical studies expressing ChrimsonR in RGCs in *rd10* mice and blind macaques showed robust response to red light, maintainance of stable visual responses over time, as well as effectively transmission of signals to the visual cortex, confirming its functional integration in the retina [115,116,117]. ChrimsonR has been successfully used in clinical trials showing expression in retinal ganglion cells and the first partial visual restoration in a blind patient with advanced RP using external goggles to enhance light detection [118]. Its ability to restore meaningful visual function, combined with safety, stability, and integration with assistive devices, makes it a leading candidate for future therapies targeting late-stage retinal degeneration.

Multicharacteristic Opsin (MCO1) is a type of microbial opsin discovered in marine organisms, often algae or related protists, which exhibits multiple desirable characteristics for optogenetics. These opsins have naturally evolved to respond to diverse light environments found in marine habitats, often adapting to varying light intensities and spectra underwater. MCO1 has shown strong, stable expression in retinal bipolar cells, as well as behavioral recovery in *rd10* mice and safety in dogs [119,120]. It is currently being tested in several clinical trials (NCT04945772, NCT05417126) in patients with RP and Stargardt disease, showing significant improvements and promising results [121].

#### 3.2.2. Hyperpolarizing Opsins

Among the hyperpolarizing opsins, NpHR is a halorhodopsin from *Natronomonas pharaonis*. It is a light-gated chloride pump activated by yellow light that hyperpolarizes neurons by driving chloride ions into the cell. Co-expression of ChR2 (ON) and NpHR (OFF) in retinal ganglion cells allowed for natural-like ON/OFF signaling, mimicking healthy retinal function [122]. eNpHR, an enhanced version of NpHR, was developed to improve membrane localization and reduce cellular toxicity caused by protein aggregation. Expression of eNpHR in residual cones of RP mouse models restored visual responses via functional retinal circuits, substituting the native phototransduction cascade, restoring light sensitivity and enabling OFF responses in photoreceptor-deficient retinas [123,124].

Jaws, an engineered red-shifted halorhodopsin derived from *Halobacterium salinarum* and related archaea, is a light-driven chloride pump engineered for improved light sensitivity across a broad range of wavelengths (470–600 nm). Jaws demonstrates superior performance compared to eNpHR in *rd1* mice, enabling precise neuronal inhibition, making it ideal for restoring OFF responses in retinal circuits [86]. More recently, it was reported to be effective in macaques (in vivo) and in human retinal models (postmortem explants and iPSC-derived organoids), successfully restoring red-light-elicited hyperpolarization in cones [105,125]. It is currently one of the most promising tools for cone-targeted optogenetic therapy.

In conclusion, optogenetic therapy is rapidly progressing from bench to bedside, where microbial opsins dominate current trials. While early trials showed promise in restoring partial vision, the next steps involve improving visual quality, refining targeting strategies, and ensuring safety and reproducibility. With continued innovation and cross-disciplinary collaboration, optogenetics holds great potential to transform treatment for patients with advanced retinal degeneration.

### 3.3. Novel Gene Therapy Approaches Inspired by Marine Organisms: Ciona Intestinalis Alternative Oxidases (CiAOX)

Impairment of mitochondrial function—whether direct or indirect—is strongly implicated in the pathology of numerous NDDs. Neurons are energetically demanding cells, and their function relies on efficient energy metabolism. In neurons, glucose and/or lactate from the blood stream are used as substrates to generate ATP by oxidative phosphorylation (OXPHOS) in mitochondria [126]. OXPHOS impairment leads to energy deficit and oxidative stress: leaking electrons generate ROS, which can damage proteins, lipids, and nucleic acids, triggering further cellular dysfunction and neuronal death. Recently, the marine environment inspired a solution to bypass defects related to CIII-CIV OXPHOS components with any genetic or environmental basis: the xenotopic expression of alternative oxidases (AOXs) enzymes, present in lower organisms like plants and marine invertebrates in mammalian systems [127].

AOXs can bypass dysfunctional mitochondrial complexes III and IV by transferring electrons from ubiquinol (CoQH_2_) directly to oxygen (with water generation), without generating proton-motive force required to produce ATP. The immediate consequence is the reduction in ROS generation and the maintenance of redox homeostasis. Importantly, AOXs are activated only when Q-pool reduction level reaches 35–40%, thus acting as a molecular switch to protect cells against oxidative stress [128]. One of the most studied AOXs is derived from the marine tunicate (sea squirt) *Ciona intestinalis* (CiAOX), a close relative of vertebrates [129]. CiAOX has been successfully expressed in human cells, fruit flies, and mice, showing conditional activation only under high redox stress, thus remaining inactive in healthy cells [130,131,132]. In mouse models, CiAOX demonstrated both beneficial or harmful effects depending on the disease context [133]. In a model of GRACILE syndrome (a complex III deficiency caused by BCS1L mutation), CiAOX expression extended lifespan and prevented cardiomyopathy and kidney damage [134]. However, in a COX15 knockout model with severe muscle pathology, CiAOX worsened the condition, likely due to interference with ROS-mediated signaling required for tissue repair and adaptation [135].

Beyond these studies, overall evidence from literature proposes the cross-species CiAOX transfer as a tool to challenge disease paradigms in appropriate models, thus offering a potential cure for mitochondrial diseases [133]. However, the potential of AOX-based therapeutics have been poorly tested in neuronal disease models. Recently, Chen et al. investigated the effects of the xenotopic expression of CiAOX in the RPE of mice with electron transport chain deficiency [136]. The RPE is vital for the retina’s health, acting as the outer blood–retinal barrier, regulating immune responses, phagocytosing discarded photoreceptor outer segments and metabolically supporting photoreceptors by providing glucose. In many photoreceptor degenerative diseases, RPE dysfunction is a key factor of disease’s progression. In AMD, the RPE damage eventually results in death of both RPE cells and photoreceptors. Importantly, Chen et al. assessed the retinal consequences of RPE-restricted expression of CiAOX, with consequent stimulation of coenzyme Q oxidation and mitochondrial respiration without ATP generation [136]. Expression of CiAOX in vivo in RPE resulted beneficial in both the RPE and neuroretina, with a reduction in mTORC1 activation, hypertrophy, stress marker expression, pseudohypoxia, and aerobic glycolysis in RPE. These changes resulted in enhanced photoreceptor structure and function, as a consequence of improved glucose availability from RPE [136]. Altogether, these recent data positively support a CiAOX-based therapy for neurodegenerative diseases, and CiAOX emerges as a promising tool to rebalance mitochondrial metabolism in the RPE as well as potentially treat retinal degeneration linked to mitochondrial dysfunction in both RPE and photoreceptors. Nonetheless, the xenotopic expression of CiAOX presents several limitations, given that AOX may facilitate very different effects depending on the tissue.

## 4. Alzheimer’s Disease

The World Health Organization (WHO) estimates that over 55 million people are living with dementia worldwide, being AD the most common neurodegenerative condition cause of dementia [137]. First identified by Alois Alzheimer in 1907, AD is marked by the progressive loss of synapses and neurons, initially targeting the medial temporal lobe and eventually leading to widespread cortical atrophy.

The early AD stages are characterized by episodic memory loss, especially in recalling new information (anterograde amnesia). As AD advances, patients experience additional cognitive impairments including aphasia (language issues), apraxia (difficulty using objects), agnosia (problems recognizing people/objects), and deficits in reasoning, planning, and decision-making [138]. Though not directly fatal, AD leads to death from secondary conditions like infections or malnutrition.

The hallmark AD neuropathological features are extracellular deposits of amyloid-beta (Aβ) peptide plaques, accumulations of hyperphosphorylated tau protein forming neurofibrillary tangles, signs of vascular damage, inflammation, oxidative stress, synaptic loss, and extensive neurodegeneration [137]. The Aβ peptide derives from the proteolytic cleavage of the APP protein by the action of integral membrane proteases termed secretases. In the non-amyloidogenic pathway, α-secretase cuts APP within the Aβ region, effectively blocking the formation of amyloid-beta [139]. This produces a soluble fragment known as sAPPα—which is thought to support brain health—and a second product, known as C83, which is later cleaved by γ-secretase, releasing a harmless p3 peptide and the APP intracellular domain (AICD). On the other hand, in the amyloidogenic pathway, the β-secretase (BACE1) cuts APP at a different site, generating the fragments sAPPβ and C99. γ-secretase then processes C99, resulting in the release of Aβ peptides [139]. Depending on the specific γ-secretase cleavage site, the two most frequent forms of Aβ peptide can vary from 40 (Aβ40) to 42 amino acids (Aβ42) in length, with Aβ40 being the most common species, and Aβ42, the less common but the more prone to aggregation [140]. Many studies have addressed the physiological functions of APP and Aβ peptide; however, little is known yet. Importantly, Aβ monomers can assemble into various soluble oligomeric forms, such as dimers, trimers, and larger aggregates (oligomers and protofibrils). Further assembling of oligomers produces insoluble aggregates named mature fibrils, core component of the amyloid plaques. Despite the accumulation of amyloid plaques in the extracellular space of the AD brain representing a key pathological hallmark, they are considered less toxic than soluble oligomers, which highly contribute to synaptic dysfunction and neuronal death [141].

Depending on the age of onset, two major types of AD are generally differentiated: Early-Onset AD (EOAD) and Late-Onset AD (LOAD). EOAD occurs before age 65, it is often familial (Familial AD, FAD) and caused by rare, highly penetrant mutations in *APP*, *PSEN1* and *PSEN2* (encoding the catalytic subunits of γ-secretase) [142]. These mutations typically increase amyloid-beta (Aβ) production or shift the Aβ42/Aβ40 ratio towards the more toxic Aβ42 form. Dysregulated expression of these genes might explain around 5–10% of early-onset AD diagnosed cases AD. Mutations like the “Swedish” double mutations (K670M, N671L) and “London” mutation (V717I) in *APP*, and gene mutations in the two *PSEN* genes contribute to abnormal Aβ processing, supporting the Amyloid Hypothesis of AD pathogenesis, as stated below [142].

In contrast, LOAD is the most common form of the disease, typically starting after the age of 65, and usually being sporadic (Sporadic AD, SAD). It has a strong genetic component, with the APOE ε4 allele being the major genetic risk factor. One allele increases AD risk by 2–4 fold, whereas two ε4 alleles increase the risk up to 12 fold. Importantly, APOE ε4 may impair Aβ clearance and promote toxic Aβ oligomer formation. Although EOAD accounts for only 1–6% of cases, it has been critical for uncovering AD’s molecular underpinnings. In contrast, LOAD, representing ~95% of cases, results from multiple low-penetrance genetic variants and environmental factors (age, education, physical activity, and comorbidities, like diabetes). Recent Genome-Wide Association Studies (GWAS) have identified additional susceptibility genes for LOAD, including *CLU*, *PICALM*, *CR1*, *BIN1*, *ABCA7*, *EPHA1* [142]. Despite the different genetic origins, both EOAD and LOAD share the same neuropathological features, including amyloid plaques and neurofibrillary tangles accumulation.

Alongside plaque formation, NFTs—composed of hyperphosphorylated tau protein—represent another key histopathological hallmark of AD. Under normal conditions, tau mediates the microtubules stabilization; however, when hyperphosphorylated, tau aggregates into paired helical filaments, forming tangles [143]. According to the amyloid cascade hypothesis, postulated in 1992 by Hardy and Higgins, the abnormal production of Aβ is the initial step in triggering the pathophysiological cascade that eventually leads to AD [144]. Aβ deposition and amyloid plaque formation are described as the main processes responsible for neuronal death, while other neuropathological hallmarks (tau pathology, neurofibrillary tangles, vascular damage, and neuroinflammation) are consequences rather than causes of the disease [145]. Despite the amyloid cascade being the first hypothesis to explain the development of AD, more hypothesis have been postulated [145,146]. The Tau hypothesis links neuronal and synaptic loss to the abnormal tau protein (e.g., hyperphosphorylation, aggregation into neurofibrillary tangles) [147].

According to the cholinergic hypothesis, AD is caused by reduced synthesis of the neurotransmitter acetylcholine; therefore, the loss of cholinergic neurons in the basal forebrain leads to cognitive deficits. However, this theory does not explain the underlying cause, and cholinergic drugs (e.g., cholinesterase inhibitors like donepezil) offer only symptomatic relief [148]. Instead, the glutamatergic hypothesis suggests that the dysfunction of NMDA receptors—specifically GluN2A and GluN2B subunits—leads to synaptic failure and excitotoxicity, often induced by amyloid-beta [149]. Early treatments (e.g., cholinesterase inhibitors like donepezil) target this system. However, this theory does not explain the primary cause of glutamatergic neuron loss. The serotonergic hypothesis proposes that dysregulation of the brain’s serotonin (5-hydroxytryptamine, 5-HT) system plays a significant role in the development and progression of AD, particularly in relation to cognition, mood, neuroinflammation, and neuroplasticity [150]. According to the oxidative stress hypothesis, the increase in reactive oxygen species induces DNA damage, leading to progressive cell death and contributing to amyloid and tau pathology. However, it is hard to determine whether oxidative stress is a primary trigger of neuronal dysfunction or a secondary effect [145]. Similar criticisms concern the mitochondrial (or metabolic) hypothesis, according to which, mitochondrial dysfunction impairs energy metabolism, triggering oxidative stress and metabolic dysfunctions. The neuroinflammation hypothesis is supported by the typical increase in inflammation markers found in AD and links the pathology to chronic activation of the brain’s innate immune system: microglia become activated in response to plaques, releasing inflammatory cytokines that damage neurons [145]. However, it remains unclear if inflammation is a cause or a consequence of plaque and tangle formation.

Since the 1990s, the amyloid cascade hypothesis has driven drug development efforts aimed at reducing beta-amyloid to treat AD, including monoclonal antibodies targeting different forms of beta-amyloid [146]. Despite multiple failures in clinical trials, three monoclonal antibodies—Aducanumab, Lecanemab, and Donanemab—have received FDA approval in the U.S. for early-stage AD [147]. However, the long-term efficacy and safety of these compounds remain to be fully validated. As a result, the search for safer and more effective treatments for AD continues to be a significant and urgent challenge.

### 4.1. Marine Drugs in Alzheimer’s Disease

Recent therapeutic strategies for AD have increasingly turned to marine drug discovery. Marine organisms produce a wide array of bioactive compounds, with unique chemical structures and diverse biological activities. These substances range from small peptides and enzymes to complex secondary metabolites, many of which exhibit neuroprotective, anti-inflammatory, antioxidant, and anti-amyloidogenic effects [10,148,149,150] (Figure 1 and Figure 2; Table 2).

#### 4.1.1. Fucoxanthin

In a previous section of this review, we have already described carotenoids (fucoxanthin and astaxanthin) as potential therapeutic agents in RDDs due to their antioxidant and anti-inflammatory properties. In AD, fucoxanthin was found to reduce the formation of Aβ fibrils and oligomers in vivo and improve cognitive function in Aβ oligomer-injected mice by enhancing brain-derived neurotrophic factor (BDNF) expression and increasing choline acetyltransferase (ChAT)-positive regions in the hippocampus [151]. Moreover, fucoxanthin has been reported to reverse scopolamine-induced cognitive impairment in vivo. In this study, authors also show that fucoxanthin inhibits acetylcholinesterase activity in vitro, acting as a non-competitive acetylcholinesterase inhibitor and counteracting the enzyme activity alterations induced by scopolamine [152]. Thus, fucoxanthin exhibits potential therapeutic efficacy for the treatment of AD by acting on multiple targets. Similarly, research by Pangestuti et al. [153] demonstrated that fucoxanthin alleviates Aβ-induced oxidative stress in microglial cells, suggesting its potential role in AD treatment. Recently, Jiang et al. (2025) demonstrated that long-term fucoxanthin administration significantly prevents cognitive deficits and Aβ-related neuroinflammation in APP/PS1 transgenic mice. The mechanism involves the inhibition of Nogo-A, leading to reduced activation of Rho-associated protein kinase 2 (ROCK2) and nuclear factor kappa-B (NF-κB) pathways [154].

#### 4.1.2. Astaxanthin

Babalola et al. showed that astaxanthin enhanced the clearance of misfolded proteins in primary porcine brain capillary endothelial cells by inducing autophagy and Aβ clearance, and reducing the secretion of inflammatory cytokines and microglia activation [155]. The effects of free astaxanthin (F-AST) and docosahexaenoic acid-acylated AST monoester (AST-DHA) on ganglioside metabolism has been tested in the cortex of APP/PS1 mice, revealing that astaxanthin supplementation improves cognitive function and reduces Aβ deposition [156]. Notably, the study suggests that ganglioside homeostasis in the AD mouse cortex might be a critical target in AD therapy. Recently the effects of docosahexaenoic acid-acylated astaxanthin monoester (AST-DHA) has been studied both in vitro and in vivo AD models [157]. The results indicated that AST-DHA reduced toxic amyloid-beta (Aβ42) levels and alleviated autophagy dysfunction in SH-SY5Y cells. In APP/PS1 mice, dietary supplementation of AST-DHA increased hippocampal and cortical autophagy, mitigated cerebral Aβ and phosphorylated tau deposition, and improved neuronal function. Mechanistically, AST-DHA restored autophagy by activating the ULK1 signaling pathway and alleviating mitochondrial stress, supporting its potential as an autophagy inducer for maintaining brain health.

#### 4.1.3. Cerebrosides

Cerebrosides, a class of neutral glycosphingolipids, are key compounds in the metabolism of sphingolipids and are abundant in the nervous systems of marine organisms, particularly echinoderms. Recent studies have highlighted their potential neuroprotective effects in AD. A study demonstrated that cerebrosides from sea cucumbers improved cognitive function in rats with Aβ42-induced deficits [158]. Administration of these cerebrosides ameliorated memory impairments and reduced neuronal damage by modulating apoptotic pathways and enhancing synaptic plasticity through the BDNF/TrkB/CREB signaling pathway [158]. The neuroprotective effect of dietary sea cucumber cerebrosides (SCGs) was tested in senescence-accelerated mouse prone 8 (SAMP8) mice, where it ameliorated the Aβ accumulation and learning and memory deficits [158]. Song et al. explored the effects of SCGs on sphingolipid metabolism and AD in mice [181]. Using behavioral tests, ELISA, and mass spectrometry, researchers found that SCG supplementation improved spatial memory in AD-model mice (SAMP8). AD mice showed altered sphingolipid profiles, including increased ceramides and cerebrosides and decreased sulfatides in the hippocampus and cortex. SCG intake modulated these lipid changes, particularly in the hippocampus. The findings suggest that SCGs can influence brain sphingolipid composition and may have therapeutic potential in mitigating AD-related cognitive and biochemical impairments.

#### 4.1.4. Fucoidan

Fucoidan, described above as potential therapeutic agents RDDs, has garnered attention for its potential neuroprotective effects in AD, due to its multifaceted mechanisms, including antioxidant properties, modulation of Aβ aggregation, mitochondrial protection, and regulation of apoptosis pathways. Studies in *C. elegans* and *Drosophila melanogaster* have shown that fucoidan alleviates AD-like symptoms, including motor deficits and cognitive impairment [159]. In particular, supplementation of fucoidan alleviated the paralyzed phenotype induced by Aβ in a transgenic *C. elegans* AD model, by reducing Aβ toxicity and decreasing Aβ-induced production of ROS. Wei et al. demonstrated protective effects of fucoidan against Aβ-induced toxicity, both in vitro in PC12 cells, and in vivo in AD model mice, where fucoidan improved learning and memory impairment induced by D-galactose subcutaneous injection [160]. Altogether these approaches indicated that fucoidan acts by inhibiting the release of cytochrome c from the mitochondria and caspase-mediated apoptosis. Moreover, it improved the antioxidant activity of superoxide dismutase (SOD) and glutathione (GSH). Finally, fucoidan regulates the cholinergic system, by acting on the activity of acetylcholine (ACh) and choline acetyl transferase (ChAT), as well as that of the acetylcholine esterase (AChE).

#### 4.1.5. Phlorotannins

Phlorotannins are polyphenolic compounds abundant in brown seaweed like *Ecklonia cava*. They offer a multifaceted approach to AD therapy, targeting key pathological processes such as cholinergic dysfunction, oxidative stress and inflammation [182]. Phlorotannins such as phlorogucinol, eckol and dieckol have demonstrated inhibitory effects on acetylcholinesterase (AChE) and butyrylcholinesterase (BuChE), enzymes responsible for the breakdown of acetylcholine [161]. By inhibiting these enzymes, a phlorotannins mixture composed by phlorotannin and fucoidan (P4F6) has been reported to help in maintaining higher levels of acetylcholine in Aβ-mice, thereby alleviating cognitive deficits. The same study showed that, at the molecular level, P4F6 attenuated Aβ-induced oxidative stress and enhanced mitochondrial function [161]. In addition, P4F6 regulated tau hyperphosphorylation by regulating the protein kinase B (Akt) pathway, and promoted the expression of brain-derived neurotrophic factor (BDNF) in brain tissue [161]. Another mechanism of action demonstrated for phlorotannins is the inhibition of Aβ peptides production and aggregation [162,163,164]. For example, dieckol from *E. cava* inhibited BACE1 activity, leading to reduced Aβ synthesis and oligomerization, thereby protecting SweAPP N2a cells from Aβ-induced cytotoxicity [164]. Phloroglucinol, a polyphenol component of phlorotannin, has been shown to regulate synaptic plasticity in an AD mouse model following stereotaxic injections in brain [165]. Moreover, oral administration of phloroglucinol for 2 months attenuated the impairment in cognitive function observed in 5X familial AD (5XFAD) mice, causing a reduction in the number of amyloid plaques and in the protein level of BACE1, and reducing microglia activation and inflammation in the brain [165,166]. While preclinical studies provide compelling evidence of their neuroprotective effects, further research is required to overcome challenges related to the low oral bioavailability of phlorotannins and their limited ability to cross the BBB.

#### 4.1.6. Sodium Oligomannate

Sodium oligomannate (GV-971) is a marine algae-derived mixture of oligosaccharides isolated from *Ecklonia kurome*, developed by Shanghai Green Valley Pharmaceuticals for the treatment of AD. GV-971 received its first approval in November 2019 in China for the treatment of mild to moderate AD to improve cognitive function. GV-971 stands out in Alzheimer’s research due to its multi-target mechanism of action [183]. It primarily acts on the gut–brain axis, targeting neuroinflammation and dysbiosis, differently from traditional amyloid- or tau-targeting drugs. GV-971 rebalances gut microbiota, reducing the abundance of bacteria that promote systemic inflammation and reducing immune cell infiltration into the brain [167].

#### 4.1.7. Gracilins

Gracilins are a group of bioactive diterpenoid marine compounds, primarily isolated from the marine sponge *Spongionella gracilis*. They have reported neuroprotective and antioxidant properties, making them interesting candidates for neurodegenerative disease research, including AD. In particular, their action has the effect of counteracting oxidative stress in neurons [168], support mitochondrial function [169] and suppress inflammatory pathways [168]. However, overall research on gracilins is preclinical and there is no clinical trial data produced in AD patients.

#### 4.1.8. Homotaurine

As discussed above, homotaurine (also known as tramiprosate) is a naturally occurring amino sulfonate compound originally found in red algae. It has been extensively studied for its potential role in treating AD, particularly for its anti-amyloid properties. Its main therapeutic mechanism of action is the inhibition of neurotoxic Aβ oligomers formation by binding Aβ peptides and preventing their aggregation [184]. This stabilization reduces the neurotoxic effects of Aβ and may preserve synaptic integrity [184]. Due to its high similarity with the inhibitory neurotransmitter GABA, homotaurine might also modulate neurotransmission, potentially contributing to calming hyperactive neural circuits in AD. A Phase III Clinical Trial (ALPHASE) has been performed to assess the clinical efficacy, safety, and disease-modification effects of homotaurine (Alzhemed(TM)) in mild-to-moderate AD [170]. The study did not show significant improvement in primary cognitive outcomes across all participants. However, a genetically defined subgroup of patients (ApoE4 carriers) showed benefit, with slower cognitive and functional decline. After the ALPHASE trial, homotaurine was not approved as a drug for Alzheimer’s treatment due to modest overall efficacy. However, its potential remains of great interest, especially in early-stage or at-risk populations, such as Apo ε4 carriers. Moreover, Alzhemed and other prodrugs of homotaurin such as Vivimind, have been marketed as a neutraceutical supplement to manage memory loss.

#### 4.1.9. Spirolides

Spirolides, particularly 13-desmethyl spirolide C (SPX), are cyclic imine toxins produced by marine dinoflagellates such as *Alexandrium ostenfeldii*. They are primarily known for their neurotoxic effects in marine organisms and recent research explored their potential therapeutic applications in AD [185,186]. The mechanism of action of spirolides is mainly based on the interaction with cholinergic receptors, including muscarinic and nicotinic acetylcholine receptors. Specifically, 13-Desmethyl spirolide-C treatment in 3xTg-AD neurons resulted in positive modulation of cholinergic signaling pathways in vitro [171]. Moreover, the same treatment decreases in the expression levels of kinases involved in tau phosphorylation [171].

#### 4.1.10. Chitosan

Chitosan is a marine-origin polysaccharide obtained from the deacetylation of chitin, the main component of crustacean exoskeleton. It has received attention in AD research due to its neuroprotective properties. Chitosan has been shown to control the level of Aβ by modulating the activity of BACE-1 [172]. Moreover, in vivo studies showed that chitosan can reduce cognitive deficits: in an amyloid-β42-induced rat model, oral administration of chitosan significantly improved learning and memory performance. This improvement was associated with reduction in neuronal apoptosis, decrease in oxidative stress markers, increased activities of antioxidant enzymes and mitigation of neuroinflammation [173].

#### 4.1.11. 9-Methyl-Fascaplysin

9-Methylfascaplysin is a synthetic derivative of the marine alkaloid fascaplysin, initially extracted from the Fijian sponge *Fascaplysinopsis* sp. It showed promising neuroprotective effects in preclinical studies, by acting through different mechanisms. Molecular dynamic simulations indicate that 9-methylfascaplysin binds to negatively charged residues of Aβ42, preventing peptides aggregation, thus leading to reduce neurotoxic Aβ species [174]. The same study showed that Aβ modified by nanomolar 9-methylfascaplysin reduced neuronal toxicity in comparison with typical Aβ oligomer in SH-SY5Y cells. Le et al., corroborated these findings in vivo, showing that low concentrations of 9-Methylfascaplysin significantly prevented cognitive impairment in APP/PS1 transgenic mice. Several beneficial effects at molecular and cellular levels in these mice were observed, including reduction in Aβ plaques, modulation of Tau pathology and decreased neuroinflammation [175].

#### 4.1.12. Rifampicin

Rifampicin, a semisynthetic polyketide antibiotic originally derived from *Amycolatopsis rifamycinica*, has been identified in marine organisms, particularly marine actinobacteria such as *Salinispora*, a species isolated from marine sponges like *Pseudoceratina clavate* [187]. Rifampicin inhibits the aggregation and fibril formation of synthetic Aβ40 peptide, preventing its neurotoxicity in PC12 cells [176]. Rifampicin potential has also been tested in vivo APP- and tau-transgenic (Tg) mice: oral administration reduced Aβ and tau oligomers in the brain and improved the cognition of the mice [177]. However, rifampicin presents a limited blood–brain barrier (BBB) permeability, posing an efficacy challenge for AD treatment. To enhance brain delivery, alternative administration routes have been tested, including intranasal administration; a preclinical study found that combining intranasal administration of rifampicin with resveratrol improved memory and reduced Aβ oligomer-related pathologies in mouse models [178].

#### 4.1.13. Dictyostatin

Dictyostatin is a marine-derived macrolide isolated from the sponge *Spongia* sp. in the Maldives, which emerged as a promising candidate in AD research due to its microtubule-stabilizing activity. Dictyostatin administration in PS19 Tau mouse model resulted in improved microtubule density, reduced axonal dystrophy, and decreased tau pathology. Additionally, there was a trend toward increased hippocampal neuron survival, indicating potential therapeutic benefits [179]. However, despite its efficacy, dictyostatin has shown evidence of dose-limiting peripheral side effects, likely due to gastrointestinal complications [179].

#### 4.1.14. Anabaseine and GTS-21

Anabaseine is a naturally occurring alkaloid toxin produced by marine nemertean worms. Its synthetic derivative, GTS-21, has received attention for its potential therapeutic effects in AD due to its selective agonist activity on the α7-nicotinic acetylcholine receptor (α7 nAChR) [180]. GTS-21 agonistic activity has been tested in clinical trials, evaluating its safety by daily administration in individuals with probable AD for 18 days. Early studies indicated that GTS-21 is well-tolerated and may offer cognitive benefits, including memory and cognition enhancement activity. However, further research is necessary to confirm its clinical utility in AD.

### 4.2. Simple Model Systems from the Sea: A Challenge in Understanding Alzheimer’s Disease

Experimental models are crucial for advancing our understanding of AD pathogenesis and for conducting preclinical testing of potential therapies. To date, most experimental models have been animal-based, particularly transgenic mice engineered to express human genes associated with amyloid plaque formation (such as human *APP* and *PSEN1*) and neurofibrillary tangles (via human *MAPT* expression) [188,189]. Moreover, non-human primates are considered the model organisms that most closely recapitulate human disease profiles, offering a closer genetic, physiological and psychosocial resemblance to humans than rodents. Other organisms, including invertebrates like *Drosophila melanogaster* and *Caenorhabditis elegans*, as well as vertebrates like zebrafish and companion model systems, have also been used [190,191]. These models have significantly advanced our understanding of disease mechanisms that could not be studied directly in humans and remain critical for experimental hypothesis testing. However, no animal model perfectly recapitulates the full spectrum of human AD.

A current trend in research is the increase in marine invertebrate models that are fully amenable to genetic manipulation and equipped with the same modern molecular and genetic tools used in well-established terrestrial model organisms. Remarkably, the biological functions and organization of the CNS are closely linked to the evolutionary history of animals.

#### 4.2.1. Tunicates

Molecular phylogenetic studies have shown that tunicates, such as ascidians, are the closest living relatives of vertebrates [192]. The solitary tunicate *Ciona intestinalis* has been presented as a novel invertebrate model system for studying AD pathogenesis [193]. Virata et al. showed that, in contrast to traditional invertebrate models such as *Drosophila melanogaster*, which lack a functional Aβ sequence and a β-secretase ortholog, the *Ciona intestinalis* genome contains orthologs of all major AD-related genes [193]. *Ciona intestinalis* larvae can correctly process human APP695, producing Aβ peptide able to aggregate into amyloid-like plaques. Plaque deposition is markedly enhanced in larvae expressing a familial AD-associated APP695 variant. Furthermore, targeted expression of Aβ in the nervous system disrupts normal larval behavior during substrate attachment. Importantly, treatment of transgenic larvae with the anti-amyloid compound 3 amino-1-propanesulfonic acid (3-APS), leads to a decrease in plaque formations, and an improved larval attachment. Authors proposed *Ciona intestinalis* as novel animal platform to investigate APP processing, Aβ aggregation and behavioral outcomes, and to perform screening for the identification of therapeutic solutions [193]. Unfortunately, studies in this direction have not been continued.

*Botryllus schlosseri* is a colonial tunicate that undergoes a unique weekly cycle of adult resorption and regeneration from buds [194]. They possess multiple brains simultaneously during their colonial phase, providing a natural model of aging and neural degeneration. Remarkably, *Botryllus schlosseri* genome contains orthologs of many genes involved in AD, including those related to APP processing. Recent research shows that during the weekly budding cycle of the colony (asexual reproduction), *Botryllus schlosseri* brain experiences a reduction in the number of neurons, prior to cell death and phagocytic removal [195]. This neurodegeneration is also associated with a decline in behavioral responses and significant changes in the expression of 73 genes homologous to those associated with neurodegenerative diseases in mammals. Additionally, *Botryllus schlosseri* seems to accumulate mutations in its genes similarly to humans. Anselmi et al., took advantage from 20 year-old colonies and showed that they had significantly fewer neurons and reduced behavioral responses in comparison with younger colonies. At the molecular level, these phenotypes corresponded to two distinct, yet interconnected, neurodegenerative processes [195]. Overall, this study highlighted *Botryllus schlosseri* as a new model to explore evolutionary conserved mechanisms underlying AD, with high potential for testing therapeutic strategies targeting neurodegenerative processes.

#### 4.2.2. Echinoderms

Sea urchins (Echinoidea) have long been model organisms in developmental biology, and their role in neurobiology is increasingly recognized. They hold a key phylogenetic position as the only non-chordate within the deuterostome clade, making it particularly valuable for comparative studies. Therefore, insights obtained by studying sea urchin embryogenesis can be meaningfully extrapolated to higher deuterostomes, including mammals. Sea urchins such as *Paracentrotus lividus* and *Sphaerechinus granularis*, have emerged as valuable model organisms for studying AD, offering insights into Aβ toxicity, neurodevelopmental processes, and potential therapeutic interventions [196,197,198].

First studies in sea urchins showed that exposure of *Sphaerechinus granularis* embryos to 0.1 μM Aβ42 induced skeletal damage, ectodermal cell accumulation, and underdeveloped larval arms, while higher concentrations (0.2–0.4 μM) produced stage-dependent defects [196]. Importantly, lipid-permeable analogs of acetylcholine, serotonin, and cannabinoids reduced these toxic effects, with the acetylcholine analog offering the strongest protection [196]. These findings support the use of sea urchin embryos as a rapid, effective model for studying Aβ42 toxicity and screening neuroprotective compounds. Moreover, the protective role of acetylcholine analogs also reinforces the therapeutic potential of cholinergic-based interventions in AD [196]. The same group also examined the effects of exogenous APP(96-110) and Aβ42 on sea urchin embryo and larvae development [199]. Both peptides caused developmental abnormalities, which were reduced or prevented by neurotransmitter analogs for acetylcholine, serotonin and cannabinoids, highlighting APP developmental role and its interaction with neurotransmitter systems that act as morphogens in both sea urchins and mammals [199].

Similarly, another study performed in *Paracentrotus lividus* supports the use of sea urchin embryos as a simple and effective model for studying Aβ42 cytotoxicity and identifying molecular pathways involved in AD. Researchers identified an APP-related antigen in sea urchins, named PlAPP, which is processed into a ~10 kDa polypeptide after the gastrula stage [197]. Moreover, they showed that exposure of sea urchin embryos to both oligomeric and fibrillar forms of Aβ42 triggered apoptosis. Importantly, treatment with a caspase inhibitor prevented apoptosis and restored normal morphology [197]. *Paracentrotus lividus* embryos have also been used to test the toxicity of recombinant β-amyloid prefibrillar oligomers, by using a novel assay [200]. Results showed that small oligomers, formed under near-physiological conditions, were more harmful than larger aggregates, showing a dose-dependent impact on sea urchin morphogenesis. Overall, the findings demonstrated that sea urchin embryos are a sensitive and effective model for studying Aβ42 oligomer toxicity, and that small, soluble oligomers exert a stronger disruptive effect on early development than larger aggregates [200].

More recently, *P. lividus* has been considered as a tool to study the presenilin (PSEN) function [198]. In the sea urchin embryo, *PSEN* gene is present in unduplicated form and encodes a protein highly similar to the two human *PSEN* genes. Moreover, *PSEN* has been shown to play a pivotal role in sea urchin early development [198].

## 5. Conclusions, Challenges, and Future Perspectives

Marine ecosystems, covering more than 70% of the Earth’s surface, are home to an immense diversity of organisms that have evolved unique biochemical pathways and molecules to survive under extreme environmental conditions. Therefore, marine organisms represent a promising reservoir of bioactive compounds that may coadiuvate standard therapies, potentially enhancing the efficacy of conventional drugs and exerting synergistic or additive effects in the treatment of neurodegenerative diseases.

Contrary to many conventional single-target drugs typically synthesized in labs, many bioactive molecules coming from the marine world are characterized by their polypharmacological potential and can act in multi-target or pleiotropic therapeutics. They possess unique chemical structures and diverse biological activities, providing high level of novelty in chemical structure and mechanisms of action. Beneficial properties of marine compounds in neurodegenerative pathologies—where multiple alterations of neuronal physiology simultaneously contribute to the pathological outcome—range from antioxidant and anti-inflammatory effects to modulation of protein aggregation, synaptic plasticity and mitochondrial function. However, further studies based on both basic research and systems pharmacology are essential to clarify the underlying molecular mechanisms and support clinical translation.

An important bottleneck associated with the development of new therapeutics targeting the CNS is the difficulty of many drugs to reach the target tissue, for instance, due to the BBB and BRB, highly selective barriers that protect the CNS from foreign substances [19]. High-molecular-weight compounds are not able to cross the BBB in pharmacologically significant amounts. Moreover, passive diffusion through the BBB is prevented by the hydrophilic features of many marine bioactive compounds, such as polysaccharides (e.g., fucoidan, laminarin) and peptides. On the other hand, lipophilic compounds, such as certain marine alkaloids and terpenoids, while better suited for BBB penetration, may suffer from rapid metabolism, low water solubility, or off-target effects [193]. Interestingly, the few drugs able to cross the BBB may “interact” with the proteins or transporters, modifying their properties or the efficiency of their action. This may explain the differences in neuroprotective effects between preclinical and clinical trials of many neuroprotective compounds [194]. Oral administration does not represent a solution due to the poor absorption of many marine-derived drugs, degradation by gastric enzymes, and first-pass hepatic metabolism. Therefore, as 98% of all small molecules do not cross the BBB, neurodegenerative disorders could highly benefit from improved strategies of drug delivery. The scientific community is actively working on this direction using different approaches, by: (1) studying novel drug delivery strategies, such as nanoparticle-based systems (e.g., liposomes, solid lipid nanoparticles and polymeric nanoparticles); (2) developing specific chemical modifications (e.g., PEGylation or conjugation with targeting ligands) to improve both stability and brain uptake; (3) exploring novel administration routes, such as the intranasal administration to bypass the BBB via the olfactory and trigeminal pathways. As an example, astaxanthin–loaded Nanostructured Lipid Carriers NLCs administered by intranasal route has been shown to promote drug distribution into the brain and significantly improve the astaxanthin therapeutic properties in murine models [195].

Remarkably, most of the studies presented in this review were conducted using in vitro systems or murine models, while a comprehensive understanding of their therapeutic relevance in humans requires further preclinical validation and well-designed clinical trials. The main limitation for translational research in the field of the marine organisms-derived bioactive drugs is the high production costs for these molecules [201]. Marine organisms are often sourced from remote environments, requiring expensive equipment and trained personnel. Seasonal variability, environmental regulations, and biodiversity protection laws further increase the cost and complexity of sustainable sourcing. Moreover, many marine species produce bioactive substances in low amounts, requiring large biomass volumes or complex synthetic methods of production, and hampering the standardization of the isolation and purification procedures [201]. Total synthesis or biosynthetic engineering represent alternatives similarly priced due to the multi-step process of chemical synthesis often required to synthesize these compounds. Importantly, heterologous expression systems and marine microorganism fermentation offer cost-saving potential, but require significant upfront investment in research and development [202].

Overall, the accurate characterization of many neuroprotective marine compounds remains stuck in the preliminary stage of in vitro testing, failing to reach the requirements for the sustainable implementation of in vivo pre-clinical and clinical trials [203]. Clinical trials for neurodegenerative diseases are among the most expensive and lengthy studies, due to the slow progression of symptoms and the need for long-term evaluation. Consequently, toxicology studies, pharmacokinetic profiling, and clinical trials require extensive investment, especially in the field of neurodegeneration. Importantly, innovations in synthetic biology, omics integration, AI-based systems of screening, and testing, bioprocess optimization, and drug delivery systems may reduce costs over time.

In this review, we aim to highligth that the huge potential of marine organisms in the fight against the neurodegeneration is not limited to provide novel drugs for therapeutic proposal, since marine organisms are also emerging as alternative model sistems to understand the molecular mechanisms underlying neurodegenerative conditions, such as AD, as well as to provide useful platforms for testing new drug candidates. Moreover, molecules coming from the sea world, such as the tunicate AOX, are inspiring novel gene therapy solutions to fight neurodegeneration, while novel marine microbial opsins are already being considered for therapeutic applications based on optogenetics.

Finally, strategic partnerships between academia, biotech companies, and government agencies will be essential to bring to the light the translational interest of many molecular strategies coming from marine organisms, as well as to accelerate the development of marine-derived therapeutics for human use and make them more accessible and economically sustainable.

## Figures and Tables

**Figure 1 marinedrugs-23-00315-f001:**
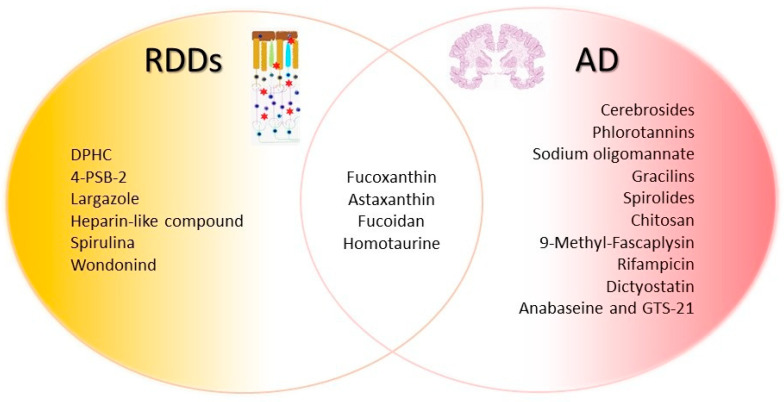
Marine organisms provide beneficial bioactive molecules that can modulate neuronal function in pathological contexts. A schematic representation of neuroprotective compounds from marine organisms in Retinal Degenerative Diseases (RDDs) and Alzheimer’s Disease (AD).

**Figure 2 marinedrugs-23-00315-f002:**
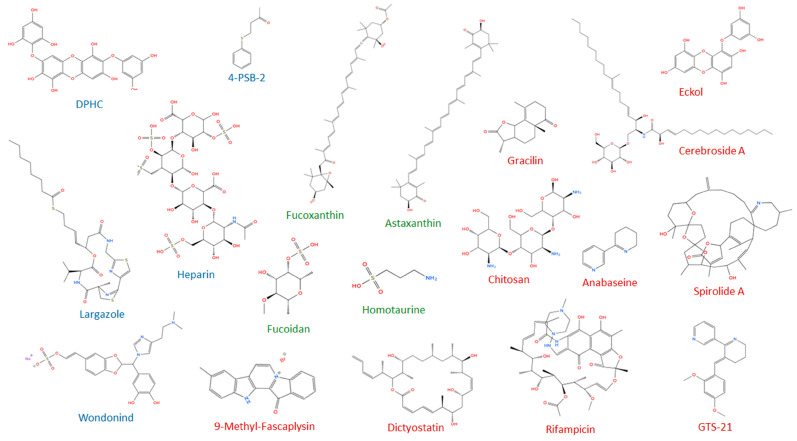
Chemical structure of marine compounds with neuroprotective effects against Retinal Degeneration Diseases (RDDs, in blue), Alzheimer’s disease (AD, in red) or both neurodegenerative diseases (in green). Sodium oligomannate (GV-971) is not shown because it consists of a mixture of a marine algae-derived oligosaccharides. Heparin is shown instead of Heparin-like compound (modified heparin with no specified formula); Wondonin A is ishowed instead of Compound-31 (with no specified formula); Cerebroside A, Eckol, and Spirolide A are shown as representative molecules of the Cerebrosides, Phlorotannins, and Spirulides group of compounds, respectively. Chemical structures were drawn using MolView software, v2.4.

**Table 1 marinedrugs-23-00315-t001:** Compounds from marine resources with tested neuroprotective effects in RDDs. ↑ indicates increase and ↓ indicates decrease in the indicated cellular (e.g., inflammation, oxidative stress, apoptosis, etc.) or molecular process (e.g., pathway activation, protein and/or gene expression).

Compound(s)/ Organism(s)	Marine Organism	Mechanism of Action	Cell Line or Model of Disease
**Fucoxanthin**	*Hizikia fusiformis*, *Laminaria japonica*, *Sargassum fulvellum*	↓ lipid peroxidation, ↓ Aβ deposition, ↓ BACE1, tight junction disruption, protection from oxidative damage, anti-angiogenic modulation	ARPE-19 cells, photo-induced retinal damage animal model, (NaIO_3_)-induced AMD animal model [64,65]
**Astaxanthin**	*Haematococcus pluviali*, *Xanthophyllomyces dendrorhous*	↓ oxidative stress, ↓ apoptosis, ↓ inflammation, ↓ NF-κB, ↓ ROS, ↓ PI3K/Akt/Nrf2, ↓ Nrf2/Ho-1	RGCs in diabetic *db/db* mouse model of type-2 diabetes, RGC-5, 661W, (STZ)-induced diabetic rats, in vivo mouse models of acute glaucoma [66,67,68,69]
**Diphlorethohydroxycarmalol**	*Ishige okamurae*	↓ DNA damage, ↓ apoptosis, ↓ ROS, ↓ mitochondrial-dependent apoptosis	ARPE-19 [70]
**Fucoidan**	*Fucus vesiculosus*, *Undaria pinnatifida*	↓ VEGF, ↓ EMT, ↓ HIF1α, ↓ angiotensin, ↑ Sirtuin1, ↓ NLRP3 activation	ARPE-19, primary porcine RPE, as well as RPE/choroid perfusion organ cultures, RPE cells, in vivo models of PVR, streptozotocin- induced diabetic mice, mRECs, mice retina [71,72,73,74]
**4-(Phenylsulfanyl)butan-2-one**	*Cladiella australis*	anti-inflammatory and anti-apoptotic modulation, ↓ TNF-α, ↓ COX-2, ↓ iNOS, ↓ NF-κB	RPE, rat models of optic nerve injury [75,76]
**Homotaurine**	*Hypnea boergesenii*, *Gracilaria corticate*, *Gracilaria pygmae*	↓ apoptosis	Rat retinal primary cultures, rodent models of hypertensive glaucoma, clinical trials [77,78,79,80,81]
**Largazole**	*Symploca hydnoides*	anti-inflammatory and anti-angiogenetic, ↓ HDACs	mouse model of alkali-induced corneal neovascularization (CNV), HEMC-1 [82,83]
**Heparin-like** **compound**	*Litopenaeus vannamei*	Anti-coagulant, anti-inflammatory, anti-angiogenetic haemorrhagic, ↓ VEGF, ↓ TGF-β1	ARPE-19, in vivo murine models (laser-induced choroidal neovascularization) [84,85]
**Spirulina**	*Spirulina platensis*, *Spirulina maxima*	↓ ROS,↓ superoxide dismutase2, ↓ heme oxygenase-1, ↓ Nrf2	A2E-Laden ARPE-19, BALB/c mice (blue light-induced retinal damage), BALB/cAJcl mice (photostress-induced retinal damage) [86,87]
**Wondonins** **(or compound 31)**	*Poecillatra wondoensis*, *Jaspis* sp.	anti-angiogenetic, ↓ VEGF/VEGFR2, ↓ ANGPT2	Zebrafish model of diabetic retinopathy, mouse models of choroidal neovascularization and oxygen-induced retinopathy [88,89]

**Table 2 marinedrugs-23-00315-t002:** Compounds from marine resources with tested neuroprotective effects in AD. ↑ indicates increase and ↓ indicates decrease in the indicated cellular (e.g., inflammation, oxidative stress, apoptosis, etc.) or molecular process (e.g., pathway activation, protein and/or gene expression, etc.).

Compound(s)	Marine Organism	Mechanism of Action	Cell Line or Model of Disease
**Fucoxanthin**	*Hizikia fusiformis*, *Laminaria japonica*, *Sargassum fulvellum*	↑ BDNF, ↓ acetylcholinesterase activity, ↓ Nogo-A ↓ ROCK2, ↓ NF-κB, ↓ oxidative stress, ↓ neuroinflammation	Aβ oligomer-injected mice, ICR albino mice, APP/PS1 transgenic mice [151,152,153,154]
**Astaxanthin**	*Haematococcus pluviali*, *Xanthophyllomyces dendrorhous*	↑ autophagy, ↓ mitochondrial stress, ↓ neuroinflammation, ↓ microglia activation, ↓ Aβ42, ↑ ULK1 signaling pathway	Primary porcine brain capillary endothelial cells, APP/PS1 mice, SH-SY5Y [155,156,157]
**Cerebrosides**	*Sea cucumbers*	Apoptotic pathway modulation, BDNF/TrkB/CREB regulation, modulation of brain sphingolipid composition	Senescence-accelerated mouse prone 8 (SAMP8) mice [158]
**Fucoidan**	*Fucus vesiculosus*, *Undaria pinnatifida*	Antioxidant activity, modulation of Aβ aggregation, mitochondrial protection, ↓ apoptosis, ↓ Aβ-induced production of ROS, ↓ cytochrome c release, ↓ caspase-mediated apoptosis, ↑ activity of superoxide dismutase (SOD) and glutathione (GSH), regulation of acetylcholine transferase, choline acetyl transferase and acetylcholine esterase	*C. elegans* AD model, *Drosophila melanogaster* AD model, AD mice model, PC12 cells [159,160]
**Phlorotannins** (e.g., phlorogucinol, eckol and dieckol)	*Ecklonia cava*	↑ acetylcholine levels, ↓ Aβ peptides production and aggregation, ↓ amyloid plaques, ↓ microglia activation and inflammation, ↓ Aβ-induced oxidative stress, ↑ mitochondrial function, regulation of tau hyperphosphorylation, regulation of Akt pathway, ↑ BDNF, ↓ BACE1	(Aβ)-mice, 5X familial AD (5XFAD) mice [161,162,163,164,165,166]
**Sodium oligomannate**	*Ecklonia kurome*	Acts on the gut–brain axis, targets neuroinflammation and dysbiosis, rebalances gut microbiota, reduces immune cell infiltration into the brain	Approved in 2019 in China for the treatment of mild to moderate AD to improve cognitive function [167]
**Gracilins**	*Spongionella gracilis*	Neuroprotection, antioxidant activity, ↓ oxidative stress, ↑ mitochondrial function, ↓ inflammatory pathways	SH-SY5Y, mouse primary cortical neurons [168,169]
**Homotaurine**	*Hypnea boergesenii*, *Gracilaria corticate*, *Gracilaria pygmae*	Preventing Aβ peptides aggregation	Phase III Clinical Trial (ALPHASE) [170]
**Spirolides**	*Alexandrium ostenfeldii*	Interaction with cholinergic receptors, ↓ tau phosphorylation	3xTg-AD neurons [171]
**Chitosan**	*Crustaceans*	BACE-1 modulation, ↓ apoptosis, ↓ oxidative stress, ↑ antioxidant enzyme activity, ↓ neuroinflammation	HEK293 APPswe cells, amyloid-β42-induced rat model [172,173]
**9-Methyl-Fascaplysin**	*Fijian sponge*	↓ neuroinflammation, preventing Aβ42 peptide aggregation, ↓ neurotoxic Aβ species, ↓ Aβ plaques, modulation of Tau pathology	SH-SY5Y, APP/PS1 transgenic mice [174,175]
**Rifampicin**	*Salinispora*	Inhibition of the aggregation and fibril formation of synthetic Aβ40 peptide, ↓ Aβ and tau oligomers	PC12, APP-(Tg) mice, tau-transgenic (Tg) mice [176,177,178]
**Dictyostantin**	*Spongia* sp.	↑ microtubule density, ↑ survival, ↓ axonal dystrophy, ↓ tau pathology	PS19 Tau mouse model [179]
**Anabaseine and GTS-21**	*Nemertean worms*	Selective agonist activity on the α7 nicotinic acetylcholine receptor, ↓ tau phosphorylation	PD mouse models [180]

## Data Availability

Not applicable.

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
