# Peer review of "Marine Derived Strategies Against Neurodegeneration"

_marinedrugs, 2025, doi:10.3390/md23080315_

Round 1

Reviewer 1 Report

Comments and Suggestions for Authors

The review article entitled "Marine derived strategies against neurodegeneration" by V. Toulis et al. is a very comprehensive review of marine drugs that have neuroprotective therapeutic potential. The article is well organized and structured so as to present a brief overview of neurodegeneration first, and then to focus on retinal neurodegeneration and present marine drugs in retinal neurodegeneration. The next section introduces Alzheimer's disease and the marine-derived compounds that have a therapeutic potential. The last concluding section provides a nice recap with suggestions for future perspectives.

Overall the article is appropriate for publication in Marine Drugs and will certainly draw the interest of its readers. However, I cannot suggest its acceptance before the authors add the chemical structures of the compounds presented. I reckon that a comprehensive review of any kind of "drugs" should be accompanied by the chemical structure of the compounds. Another minor suggestion is to recolor figure 1 so as its outline to be visible, considering that the three compounds in the middle are not shown clearly at the intersection. Finally, the up/down arrow symbols used in the Tables 1 and 2 should be clearly described as increase/decrease of what? enzymatic activity? gene expression? signaling? On these grounds, why some mechanisms of action are provided in a descriptive way? e.g. inhibition of acetylcholinesterase and  butyrylcholinesterase (Table 2).

If the authors can address these concerns, I would be happy to consider a revised version of the manuscript for publication in Marine Drugs.

Author Response

We thank the reviewers for their revision and all the suggestions to improve the quality of our review. We have carefully considered all their recommendations, revised the English language and expressions, introduced the molecular structure for each compound, and changed the colours and other details in figures and Tables for a much better comprehension of the readers. Please find our detailed answer to the specific points raised by the reviewers below.

Author's Reply to the Review Report (Reviewer 1)

Comment 1: I cannot suggest its acceptance before the authors add the chemical structures of the compounds presented. I reckon that a comprehensive review of any kind of "drugs" should be accompanied by the chemical structure of the compounds.

Response 1: We thank the Reviewer for his/her comment. We totally agree about the importance of including the chemical structure of the compounds in our review, according to the comments of both the reviewers. Following the reviewers’ suggestion, we included a new figure (Figure 2) with the molecular structures of the compounds listed in the Figure 1, as well in the text. Therefore, we hope to clearly show the chemical nature of the mentioned marine neuroprotective compounds.

Comment 2: Another minor suggestion is to recolor figure 1 so as its outline to be visible, considering that the three compounds in the middle are not shown clearly at the intersection. Moreover, we moved “Homotaurine” in the middle area.

Response 2: We changed the Figure 1 following the reviewer’s suggestion and making the outlines visible. Moreover, we included Homotaurine in the group of shared compounds.

Comment 3: Finally, the up/down arrow symbols used in the Tables 1 and 2 should be clearly described as increase/decrease of what? enzymatic activity? gene expression? signaling? On these grounds, why some mechanisms of action are provided in a descriptive way? e.g. inhibition of acetylcholinesterase and  butyrylcholinesterase (Table 2).

Response 3: We thank the Reviewer for his/her comment. Descriptive information has been eliminated. We expanded the information stated in the Table legend 1 and 2 in order to clarify the meaning of the up/down arrow symbols, without compromising the synthetic and schematic structure of the tables. Further and extended information about the mechanisms of action of each compound can be found in the text and/or in the bibliography provided in the fourth column of each table.

Reviewer 2 Report

Comments and Suggestions for Authors

This review give the audience a comprehensive of marine derived strategies, including marine natural products, marine inspired gene therapies (like opsins and alternative oxidases), and marine ND models, against neurodegenerative diseases (NDs). It's an informational and well organized review to the researchers in this field. However, the following aspects should be improved.

1) The authors are suggested to provide molecular structures for the compounds in Figure 1 to clearly show the chemcal nature of the mentioned marine neuroprotectants. Please carefully check the correctness of the structures and the normality of structure drawing (using ChemDraw following the ACS1996 style)
2) Many wrong spellings exsist in the manuscript. The authors are suggested to very carefully check and correct them, such as "sveral” ( to be "several”,“exaustive” (“exhaustive”),
“dinamics”( “dynamics”),“phisiology”(“physiology”), “accellerate”(“accelerate"),“therepeutic” (“therapeutic”), “compairson”(“comparison”), “liorated”(“ameliorated”), “lipd”(lipid),“word” (line 1101,“world”), etc.
3) grammar errors, like: “The neuronal cytoskeleton... are essential”should be “is essential”; "differently" in line 956 should be "different"。
4) wrong numbering, like "2)" in line 1066 should be ""3).
5) wrong species name, for example, in table 1, "Hijikia fusiformis" should be "Hizikia fusiformis or Hizikia fusiforme or Sargassum fusiforme"; besides, all Latin name for species should be written in italicized font.
6) In table1, "Spirulina" is not suitable to be listed under the item of 'compounds'.
7) The non-unified format for reference list, for example, the inconsistent way of capitalization for paper titles.

Comments on the Quality of English Language

1) Many wrong spellings exsist in the manuscript. The authors are suggested to very carefully check and correct them, such as "sveral” ( to be "several”,“exaustive” (“exhaustive”),
“dinamics”( “dynamics”),“phisiology”(“physiology”), “accellerate”(“accelerate"),“therepeutic” (“therapeutic”), “compairson”(“comparison”), “liorated”(“ameliorated”), “lipd”(lipid),“word” (line 1101,“world”), etc.
2) grammar errors, like: “The neuronal cytoskeleton... are essential”should be “is essential”; "differently" in line 956 should be "different"ï¼›other aspects should also be checked.

Author Response

We thank the reviewers for their revision and all the suggestions to improve the quality of our review. We have carefully considered all their recommendations, revised the English language and expressions, introduced the molecular structure for each compound, and changed the colours and other details in figures and Tables for a much better comprehension of the readers. Please find our detailed answer to the specific points raised by the reviewers below.

Author's Reply to the Review Report (Reviewer 2)

Comment 1: The authors are suggested to provide molecular structures for the compounds in Figure 1 to clearly show the chemical nature of the mentioned marine neuroprotectants. Please carefully check the correctness of the structures and the normality of structure drawing (using ChemDraw following the ACS1996 style)

Response 1: We thank the Reviewer for his/her comment. We totally agree about the importance of including the chemical structure of the compounds in our review, according to the comments of both the reviewers. Following the reviewers’ suggestion, we included a new figure (Figure 2) with the molecular structures of the compounds listed in the Figure 1, as well in the text. By this way we hope to clearly show the chemical nature of the mentioned marine neuroprotective compounds.

Comment 2: 2) Many wrong spellings exist in the manuscript. The authors are suggested to very carefully check and correct them, such as "sveral” ( to be "several”, “exaustive” (“exhaustive”),
“dinamics”( “dynamics”), “phisiology” (“physiology”), “accellerate” (“accelerate"), “therepeutic” (“therapeutic”),  “compairson” (“comparison”), “liorated” (“ameliorated”), “lipd”(lipid),“word” (line 1101, “world”), etc.;

3) grammar errors, like: “The neuronal cytoskeleton... are essential”should be “is essential”; "differently" in line 956 should be "different”;

4) wrong numbering, like "2)" in line 1066 should be "3)”;

5) wrong species name, for example, in table 1, "Hijikia fusiformis" should be "Hizikia fusiformis or Hizikia fusiforme or Sargassum fusiforme"; besides, all Latin name for species should be written in italicized font;

6) In table1, "Spirulina" is not suitable to be listed under the item of 'compounds'.

Response 2: We thank the reviewer for pointing some English spelling and grammar errors that escaped our notice. We have carefully checked the English expressions and corrected all the indicated terms in this new version; we have corrected the numbering, as indicated; we have revised the name of the species, correcting and italicizing the names and genres. Finally, we modified the Table 1 by adding changing “Compound(s)” with “Compound(s)/Organism(s)” in the first line of the first column, in order to make “Spirulina” suitable to be included in the Table.

Comment 3: The non-unified format for reference list, for example, the inconsistent way of capitalization for paper titles.

Response 3: We apologize for the mistake related to the referencing style. In this revised version of the manuscript, we have carefully revised the bibliography so that the references are all presented homogeneously.

Round 2

Reviewer 1 Report

Comments and Suggestions for Authors

I can now endorse publication of the review article in Marine Drugs. The authors have revised the manuscript appropriately and now it meets the quality standards of the journal.